# The Coupling and Coordinated Development of Green Builds and Financial Development in China

**DOI:** 10.3390/ijerph19148685

**Published:** 2022-07-17

**Authors:** Zhong Fang, Hongrui Zhang, Jianlin Wang, Junbo Tong, Xiaoxiao Li

**Affiliations:** 1School of Economics, Fujian Normal University, Fuzhou 350007, China; qsz20210050@student.fjnu.edu.cn (J.W.); qsx20210013@student.fjnu.edu.cn (J.T.); qsz20210052@student.fjnu.edu.cn (X.L.); 2Department of Statistical Science, University College London, London WC1E 6BT, UK; hongrui.zhang.19@ucl.ac.uk

**Keywords:** high-tech industry innovation efficiency, DEA model, three-stage network, dynamic efficiency

## Abstract

As a labor-intensive industry with a strong industrial driving force and high-technology integration, green buildings offer some comparative advantages. Driven by the concept of green development, green buildings are ushering in a period of opportunity for integrated development among multiple fields. Therefore, this research will select the panel data of the financial industry and the green buildings industry in 2014 and 2018, respectively, in 31 provinces in China (excluding Hong Kong, Macao and Taiwan) and, through the method of factor analysis, will innovatively construct a financial industry development index and a green building Development Index for each province in China. Through the coupling coordination model, it studies the development level of the financial industry and green buildings in various provinces, in order to deeply explore the path and mechanism of coordinated development between the two. The results show that the financial industry and green buildings in the eastern coastal areas have a high level of coupling, and the coupling and coordinated development have a greater degree of correlation. The potential for coupling and coordination in Central China is developing for the better, while volatility in the Northeast and Northwest regions is relatively large. From the time dimension angle, the degree of coupling and coordination between green buildings and the financial industry in China is generally low, and in the transitional stage, from the brink of unbalanced development to a primary stage of coordinated development. Accordingly, this paper proposes that local government should pay attention to the coordination relationship between green buildings and financial industry development and formulate a coordination mechanism between their growth according to local conditions, so as to promote the correct interactive advancement of the two.

## 1. Introduction

As a pillar industry of the national economy, the construction sector has made outstanding contributions to both economic and social development. According to data released by the National Bureau of Statistics of China, the added value of the construction industry in 2019 domestically was 7090.4 billion yuan, or an increase of 5.6% over the previous year, and the growth rate increased by 0.8 percentage points. The growth rate of the added value of the construction industry was 0.5 percentage points lower than the growth rate of the GDP, but in 2019 the added value of the construction industry accounted for 7.16% of the GDP, or an increase of 0.04 percentage points over the previous year and the highest point in the past decade. The construction industry is still in a solid position for growth (Awadh & Omair, 2017 [1]) given that the world is now ushering in a fourth wave of industrial revolution as represented by artificial intelligence.

The new infrastructure represented by 5G communication technology, big data, and the Internet of Things (IoT) provides a new opportunity for the transformation and upgrading of China’s traditional construction industry (Zhao et al., 2019 [2]). In February 2017, the “Opinions of the General Office of the State Council on Promoting the Sustainable and Healthy Development of the Construction Industry” issued by central government mentioned that it is necessary to further deepen the reform of “decentralization, management and service” in the domestic construction industry and accelerate industrial upgrading of this industry. In March 2020 the Standing Committee of the Political Bureau of the Central Committee of the Communist Party of China held a meeting and proposed to accelerate the construction of new infrastructure such as 5G networks and data centers. For all countries, infrastructure construction is the foundation for the stability of an industrial society. Without modern railways, highways, electric power, and communication systems, industries cannot operate.

Looking back at the golden twenty years of the construction industry, we see that the traditional construction industry has helped spur the vigorous development of steel, building materials, cement, and other industries, which have greatly stimulated the sustained and stable growth of China’s economy. However, as this construction industry enters the later stage of development, its flood-irrigated development method not only pushes up asset prices, but also brings about problems such as industrial structure imbalance and overcapacity. These factors have adversely affected the real economy’s advancement. For the traditional construction industry, the development brought about by its rapid expansion is becoming a thing of the past, and it is facing unprecedented opportunities and challenges. Compared with traditional building development methods, green buildings have richer connotations and wider coverage (Zuo et al., 2014 [3]). Under the support of multi-field technology integration, accelerating the green transformation and upgrading of China’s construction industry can better promote the continuous growth of the economy.

In February 2019 the Political Bureau of the Central Committee of the Communist Party of China held the 13th collective study on improving financial services and preventing financial risks. It mentioned the aim of deepening the structural reform of the financial supply side and enhancing the ability of financial services to serve the real economy. Financial services’ relationship to the real economy is an important requirement for the development of the financial industry. Especially in the context of the new era, China’s economy has shifted from a high-speed growth stage to a high-quality development stage. Traditional green buildings are clearly in an urgent need of transformation and upgrading, and the requirements for financial development are getting stronger and stronger.

The relationship between China’s financial system and the real economy has undergone very profound changes in recent years. On the one hand, the interconnection of various financial market channels is improving day by day, and the social financing method is gradually changing from indirect financing to direct financing, which is manifested in the significant increase in the proportion of direct financing stock and direct financing increment. On the other hand, the traditional construction industry is asset-heavy. Under the background of global economic sluggishness and industrial transformation and upgrading, green buildings are bound to enter an era of low profit. In response to the current development dilemma faced by the construction industry, relevant departments have issued policy documents such as the “13th Five-Year Plan for the Development of the Construction Industry” and the “Opinions of the General Office of the State Council on Promoting the Sustainable and Healthy Development of the Construction Industry”, indicating the future development direction of China’s construction industry—namely, industrialization, informatization, and greenization—the connection between the “three modernizations” being “to elevate the construction industry to the level of modern industrialization”. The construction of ecological civilization has become China’s current national development strategy, and finance, as the “rainbow” of enterprise operations, can achieve coupled development with green transformation and upgrading of the construction industry. This topic deserves the attention of academia.

It is imperative to couple coordinated development between the financial industry and green buildings, thereby promoting the traditional construction industry from extensive development to high-quality low-carbon development. Under the framework of the concept of green development, this paper selects the factor analysis method to construct China’s financial industry development index and green building development index, respectively, and uses the coupling coordination model to evaluate the coupling and coordinated development level of the financial industry and green buildings in various provinces, in order to explore the interaction path and regional heterogeneity between the two. The significance of the research is mainly reflected in the following three points: first, according to the existing literature, this paper selects relevant indicators that can characterize the development degree of the financial industry and green buildings, and the factor analysis method is an innovative way to construct China’s financial industry development index and green building development index, and successfully remove several irrelevant factors from the multi-dimensional correlation variables, reducing the complexity of the model. Second, the current academic research on the development of China’s financial industry and green building is mainly based on a single view and the two are rarely combined for in-depth research. Therefore, this paper innovatively constructs a coupling coordination model to explore the relationship between the coordinated development of the financial industry and green buildings, so that the two are no longer regarded from only a single viewpoint. Third, from a macro perspective, on the one hand green buildings are high-quality assets for the financial industry and on the other hand the development of green buildings is inseparable from the support of the financial industry. This paper evaluates the coordinated development level of the financial industry and green buildings in each province, in order to find out the pathway and mechanism of coordinated development between the two.

## 2. Literature Review

Paul in the 1960s merged the words ecology and architecture into arology and put forward the famous concept of green building. In the 1970s, the outbreak of the oil crisis made people soberly realize that the history of rapid civilization development at the expense of the ecological environment is unsustainable (Tayyab et al., 2016 [4]). The construction industry, which consumes the most natural resources, must change its development model and follow the road of sustainable development. The mutual promotion between green buildings and finance is mainly reflected in the fact that capital investment promotes technological progress, and the scale of the industry can be further expanded (Ahn et al., 2013 [5]). Compared with ordinary commodities, green building products have a long service life, a large amount of value storage, and stable value preservation, which help attract more capital injection, and the two can realize interconnection and interaction (Geng et al. 2012 [6]). Finance is the lubricant of the real economy, and the transformation of green buildings to green as an emerging industry form mainly involves the concept of sustainable development. On the one hand, green finance guides funds into green environmental protection industries (Lacouture et al., 2009 [7]). At the same time, it has also continued to attract more participants to innovate and design green financial products and actively guide the flow of capital into green industries such as wind power and nuclear energy. In this process, major bodies also encouraged to participate in the formation of the concept of green consumption and green production. On the other hand, the development of green industries pays attention to ecological and environmental protection, and with the support of green finance effective financing channels are obtained, and the sustainable development of green industries can inject vitality into the development of green finance (Chan et al., 2017 [8]). Specifically, green finance and green industry interact in terms of factor supply, market orientation, and related policies, thereby forming a coupled development effect. 

The action mechanism of the financial market on green building development has long been the focus of academic research. Chen et al. (2015) [9] conducted research on the relationship between mortgage loans and green buildings and noted that the relationship between mortgage loans and housing market demand and supply is not very great, but does experience elastic fluctuations, which further show that the impact of mortgage loans on green buildings is elastic. Darko et al. (2017) [10] conducted data analysis and research on fluctuations in the construction market and the loan market in the financial industry and found that fluctuation in the former relates to development in the latter. In addition, Darko et al. (2016) [11] discussed conditions and policy recommendations for the development of national green building finance. Debrah et al. (2021) [12] established a mathematical model to analyze the close relationship between the construction market and financial stability. Using the Oxford Economic Forecasting Model to study green buildings in the UK, they found, since the 1960s, that the link between UK mortgage rates and long-term interest rates has reduced volatility in building product prices. In the short term, the interest rate mechanism is the main reason for the continued rise in the prices of construction products. 

Hoffman et al. (2008) [13] analyzed the effect of China’s financial policy on green building regulation through a specific model and stated that its macro-control policy is ineffective in some cases, such as multi-party abandonment. At the same time, they proposed that one of the more effective ways of changing the bad legal habits in the construction market is to reform the green building financing system and change the binding relationship between stakeholders. Lee et al. (2020) [14] examined the transmission mechanism of financial policy in the construction market from the interest rate and credit perspectives and combined it with the transaction mechanism of the construction market and the ecological environment of construction finance, summing up the reasons why the policy effect is not significant.

The development of the financial industry is also inseparable from the support of green buildings (Nguyen et al., 2016 [15]). As high-quality, profitable, safe assets, green buildings have always been favored as investments in the financial industry, providing a solid backing for the industry’s steady development (Li et al., 2014 [16]). Liu (2011 [17]) found that the correlation between Beijing’s green buildings and the financial industry was relatively high before 2005, but decreased year by year after 2005. This shows that, if the financial industry is highly dependent on green buildings, then it will lead to its unbalanced development, thus providing a theoretical basis for the rational adjustment of Beijing’s industrial structure. Owens et al. (2006) [18] pointed out a certain relationship between the financial industry and green buildings, and that green buildings have a greater demand for financing from the financial sector. The empirical results show that, from 2002 to 2007, green buildings have always played a role in promoting financial industry development, mainly because they increase the demand for bank financing and promote financial sector growth to a certain extent (Samer et al., 2013 [19]). Shan et al. (2018) [20] believed that the level of regional infrastructure has played a certain role in promoting financial agglomeration, but the impact of the improvement of the level of infrastructure on financial agglomeration is not as significant as that of government spending on financial agglomeration. In addition, Vatalis et al. (2013) [21] found that the construction of financial infrastructure has a significant impact on financial inclusion.

Based on the above literature, this paper presents the view that few scholars have studied the relationship between green buildings and the financial industry based on the analysis of degree of coupling coordination, whereas more attention is paid to the contribution of finance to promoting the development of green buildings. This seems to weaken the actual role of the virtual economy. If the social economy is to achieve healthy development, then it is necessary to grasp the relationship between finance and the real economy. The development mechanism of the two should be one of symbiosis and co-prosperity and not dependency. Based on the above, the research significance of this paper is as follows. First, it analyzes the internal mechanism of the coupled development of green buildings and the financial industry. Second, from the concept of green development, this paper constructs an evaluation index system for the coupling coordination between green buildings and the financial industry, uses factor analysis to determine the weights of each indicator in 2014 and 2018, and then applies the degree of coupling coordination between green buildings and the financial industry. The evaluation model analyzes the coupling and coordinated development of China’s green buildings and the financial industry in time and space distributions, offering important theoretical and practical significance for promoting the development of its financial industry and the green transformation and upgrading of green buildings.

## 3. The Internal Mechanism of the Coordinated Development of Green Buildings and the Financial Industry

Coupling as a term in physics is a concept first proposed by J.R. Pierce (Illankoon et al., 2017 [22]). Although the coupling theory originally describes the coupling law between two or more electromagnetic wave modes, it has now been widely used in the mechanism analysis of the interaction and mutual influence of two or more systems. According to the above definition, the coupling relationship between the financial industry and the development of green buildings is the relationship between the financial market and the construction industry, as well as the mutual influence and interdependence between the internal elements. In other words, it is the evolution trend of the development level of the financial industry and the development level of green buildings at a certain stage.

### 3.1. The Impact of Green Buildings on the Development of the Financial Industry

The real economy forms the foundation of financial existence and development. Under the same circumstances in terms of technical level, innovation capability, and the market environment of financial institutions, the more features and the greater the complexity of the real economy served by financial capital, the greater the space for the development of the financial industry (Wang et al., 2005 [23]). Compared with general industry, the construction industry plays a pivotal role in economic development, and its reproduction process has its own particularity, which in turn creates favorable conditions for the sound development of the financial industry (Wedding et al., 2007 [24]). First, green buildings have outstanding features such as high value and a long production cycle. Compared with general physical assets, green buildings need more and more complex support and services provided by the financial market in order to operate well, thus creating space for financial innovation. Second, although the rapid growth of the construction industry has made great contributions to economic and social development, various contradictions arise in the process of reproduction, which further prompt the reform of the financial market. Finally, the forms of financial capital involved in the reproduction of green buildings are complex, such as banks, bonds, funds, etc. Different forms inevitably have different degrees of blending or collision, which lead to many technical problems. It is necessary to rely on technological innovation in the financial industry to break through these bottlenecks.

### 3.2. The Impact of the Financial Industry on the Development of Green Buildings

Finance plays a boosting role in the process of economic and social development. Reasonable financial policies not only effectively reduce information transaction costs, but also improve the conversion rate of investment and savings, so as to promote the healthy and stable development of green buildings (Dong et al., 2021 [25]). First, the financial industry can promote the structural adjustment and optimization of the green building industry through the development of financial policies. The theory of financial constraints proposes that the government needs to formulate a series of financial policies to create opportunities for the operation of real economic sectors and financial institutions. As part of the real economy, green buildings need continuous support from financial policies in their process of development. Through financial policy implementation, the means of coordination and distribution can be used to continuously create rents (Zhang et al., 2021 [26]). Therefore, the corresponding moral hazard and adverse selection can be avoided through incentives, and eventually the target of promoting the upgrading of the green building industrial structure can be achieved. 

Second, the financial market plays an important role in the supply and demand of green buildings, by providing high-quality financing services. Because green buildings have their own characteristics in the development process, the capital required is relatively large, and many projects need the support of financial policies to be carried out smoothly. Therefore, providing high-quality financial services and targeted financial products will have a profound impact on the development of green buildings. In the process of buildings’ green transformation, financial services and financial policies need to be closely combined to form an interaction and mutual optimization relationship, so as to provide long-term and important financial policy support for their healthy and stable development. Figure 1 illustrates the coupling and the coordinated development mechanism of green buildings and the financial industry.

## 4. Selection of Evaluation Indicators and Research Methods

### 4.1. Evaluation Index Design

Based on the existing literature covering the evaluation and research of financial development levels, and combined with the characteristics of China’s financial development and data availability, the evaluation index system of financial development level is constructed from three aspects: financial scale, financial environment, and financial efficiency. Goldsmith proposed the classic financial correlation ratio indicator when measuring financial development, which is defined as the ratio of the value of all financial assets to the value of all physical assets (Milad et al., 2013 [27]). Most domestic scholars have used M2/GDP to characterize the scale of the financial industry when they study China’s financial growth. With the steady development of its financial market, the increasingly perfect securities and insurance markets, and the proportion of non-monetary financial assets increasing year by year, it is not reasonable to continue employing M2/GDP to measure China’s financial scale. Therefore, this paper includes the securities market and insurance market. Scope denotes the value of all financial assets and is taken as a secondary indicator to measure the scale of financial development, including the total stock market value of each province (autonomous region, municipality directly under the central government), total liabilities of general contracting and professional contracting green building enterprises, social financing scale, and insurance premium income. The financial environment indicators are from Zaini et al. (2015) [28] and are characterized by bank employees, various deposits, and gross domestic product of banking financial institutions in each province (autonomous region, or municipality directly under the central government). Bayero (2015) [29] defined financial efficiency as the allocation efficiency of financial resources. On this basis, this paper selects the added value of the financial industry, total capital formation, and urban and rural savings as the measurement indicators of financial efficiency. 

Wei et al. (2015) [30] noted in order to implement green construction activities that it is necessary to conserve energy, land, water, and other production factors, and at the same time, in the planning and management stages, environmental pollution problems must be avoided as much as possible. According to the latest report released by the Leadership in Energy and Environmental Design (LEED) in 2016, the green building evaluation system should include sustainable site design, improve energy efficiency, and fully ensure indoor environmental quality. The green concept runs through the design process and six aspects, including innovation. The domestic green building evaluation index has not yet formed a unified conclusion in academic circles. The most authoritative document is the Green Building Evaluation Standard GB/T50378-2019 issued by the Ministry of Housing and Urban-Rural Development, which mentions that the green building evaluation index system should include safety, comfort, convenience, energy savings, and livability.

Yang et al. (2014) [31] constructed sustainable building development indicators under the framework of Press-Status-Response (PSR) and used the fuzzy comprehensive evaluation method to reduce the dimensionality of the indicators. Based on the existing literature, this paper sets up the green development level of the construction industry in each province (autonomous region, municipality directly under the central government) from three aspects: industrial scale, industrial benefit, and ecological benefit. In addition, considering that the weight of each indicator will vary with time, the weight of each indicator in 2014 and 2018 is calculated separately. The specific indicator construction appears in Table 1. All positive indicators are selected in this paper. The data selected in this article come from the Cathay Pacific database, EPS database, and China Economic Net statistics database. Some missing values are supplemented by linear interpolation method.

### 4.2. Research Method

#### 4.2.1. Factor Analysis

The purpose of removing some irrelevant factors from multi-dimensional correlated variables is firstly to simplify complex situations and characterize the meaning of comprehensive variables with the least amount of information. Before analyzing each factor, it is necessary to forward the selected indicators to prevent inconsistency of the dimensions of each indicator, and then standardize all data. Second, it is necessary to test whether the initial variables have strong correlation. If the correlation between variables is small, then common factor variables cannot be extracted. The specific formula is as follows.

##### Data Normalization

Before factor analysis, the data are dimensionless and processed to obtain a standardized vector to avoid any heteroscedasticity. This study uses the Z-score to standardize the original data. The specific formula is as follows:(1)$$x_{ij}^{\prime }=\frac{x_{ij}-\mu }{\delta }\;$$

In the formula, $x_{ij}^{\prime }$ is the standard value of the original index; $x_{ij}$ is the original index; *μ* is the mean of the *j* index; and *δ* is the standard deviation of index *j*.

Solve the Initial Common Factor and Factor Loading Matrix

From the factor analysis method, we see that k indicators can be represented by a linear combination of n (n < k) common factors F1, F2, …, Fn. The formula goes as follows:(2)Y=AF+ε 

Here, *Y* represents the original matrix after normalization; *A* is the factor loading matrix; and *F* is the common factor matrix and is the special factor matrix. This paper applies SPSS Statistics 25.0 software to extract the principal components of the above 18 variables’ data. Table 2 and Table 3 are the explained total variance and the rotated component matrix of the observations in 2014, respectively. Table 4 and Table 5 are the explained total variance and the rotated component matrix of the observations in 2018, respectively. 

##### Data Validity Check

After data normalization, a validity test is required to confirm whether the data can be weighted using factor analysis. When KMO ≥ 0.6, the data are suitable for factor analysis. Bartlett’s sphericity test is then performed to illustrate the correlation between the indicators. The test results appear in Table 6 and Table 7. The KMO values in 2014 and 2018 are both greater than 0.6, and so factor analysis can be performed on the data.

##### Determine Variable Weights 

According to the weighting method of factor analysis, the load of n common factors F1, F2, …, Fn on the index *X_j_* (*j* = 1, 2, …, *k*) is determined as *a_ij_.* The calculation formula of the weight of each index runs as follows:(3)$$b_{ij}=\frac{\left|a_{ij}\right|}{\sum_{j=1}^{k}\left|a_{ij}\right|}\;c_{ij}=\overset{n}{\underset{i=1}{\sum }}b_{ij}\;\omega_{j}=\frac{c_{j}}{\sum_{i=1}^{k}c_{j}}$$

Here, *b_ij_* is obtained by normalizing the absolute value of Fi’s load on each variable and represents the weight of each factor to be evaluated by a group of variables; *c_j_* is obtained by the average addition of the weights of each factor; and $\omega_{j}$ is the weight of each index. Table 1 lists the calculated weights.

#### 4.2.2. Evaluation of the Comprehensive Development Level of Financial Industry Subsystem and Regional Green Building Greening Subsystem

The degree of coupling model was originally used in physics research to describe the degree of interaction between a system or elements within a system. Although the degree of coupling can reflect whether the greening level of green buildings is consistent with the development process of the financial industry, it cannot measure the degree of coupling along with coordinated development. Therefore, by establishing a regional green building–financial industry coupling coordination degree evaluation model, the coupling coordination degree is introduced to measure the synergistic development effect of the two subsystems. Among these, the coupling model expression is as follows:(4)$$C=\sqrt{\frac{f_{1}\times f_{2}}{{\left[\left(f_{1}+f_{2}\right)/2\right]}^{2}}}$$

Here, *f*_1_ represents the comprehensive development level of the financial industry subsystem; *f*_2_ represents the comprehensive development level of the regional green building subsystem; *C* is the degree of coupling between the two subsystems, and the value range of *C* is [0, 1]. The larger the *C* value, the greater the degree of correlation between the two subsystems. With the increase in the degree of coupling, the system develops from disorder to order. When *C* = 1, it indicates that the two subsystems are in the best coupling state.

In order to more truly and objectively reflect the coordinated development level of the two systems, this paper also builds a service-oriented coupling coordination model of the financial industry and manufacturing industry. The specific model is as follows:(5)$$D=\sqrt{C\times T}$$
(6)$$T=\alpha f_{1}+\beta f_{2}$$

Here, *D* is the degree of coupling coordination, where the value range is [0, 1]; *T* is the overall comprehensive evaluation index of the financial industry and regional green buildings; and *α* and *β* represent the contribution coefficients of the financial industry and regional green buildings, respectively. The development of green buildings is conducive to promoting the development of regional green buildings, which in turn further promotes the development of the financial industry. Therefore, *α* and *β* are respectively set as 0.5 in this paper.

## 5. Empirical Analysis

### 5.1. Degree of Coupling Coordination Standard

Referring to the research results of Zhang et al. (2017) [32], the degree of coupling coordination is divided into six grade intervals. Each interval represents a synergy grade, and each grade corresponds to a type of cooperation state, as shown in Table 8.

### 5.2. Analysis of Research Results

Firstly, the factor analysis method is used to determine the weights of 18 indicators. Then, on the basis of constructing the coupling coordination degree model of the financial industry–green building system, 31 provinces in China (excluding Hong Kong, Macao and Taiwan regions) are taken as the research unit, and each province’s (autonomous region’s) share is calculated as the comprehensive development index *f*_1_ and *f*_2_ of the green building subsystem and the financial industry subsystem, the coupling degree C, the development level T and the coupling coordination degree D. This paper presents an empirical analysis of the coupled and coordinated development of the financial industry–green building system in each province and municipality from three points of view. The first part divides the development status of the two industry systems into the lagging type of green building development (*f*_1_ > *f*_2_), green building development, and financial industry development through the development index of green building and financial industry in each province (autonomous region, municipality directly under the central government) in China. For synchronous type (*f*_1_ = *f*_2_) and financial industry development lag type (*f*_1_ < *f*_2_), the second part is based on the degree of development and coupling of the two subsystems, the coupling and coordinated development of green buildings, and the financial industry in 31 provinces and cities. For the overall comprehensive analysis of the status quo, the third part analyzes the spatial distribution of the degree of coupling and coordination between green buildings and the financial industry according to the degree of coupling.

#### 5.2.1. Analysis of the Comprehensive Development Level of China’s Green Buildings and Financial Industry

According to Table 9, the five provinces and cities of Jiangsu, Zhejiang, Jiangxi, Hainan, and Chongqing in 2014 were lagging in financial industry development, while the other 26 provinces were lagging in green building development. It was shown in this year that the development level of green buildings in most provinces in China lagged behind the development level of the financial industry, and that the financial industry’s service for the comprehensive development of green buildings needs to be strengthened to provide a sufficient guarantee and impetus for green transformation and development of green buildings. In 2018, the 10 provinces of Beijing, Tianjin, Hebei, Inner Mongolia, Liaoning, Jilin, Heilongjiang, Shanghai, Hainan, and Gansu were lagging behind in the development of green buildings, while the remaining provinces were lagging behind in the development of the financial industry. It can also be seen from 2014 to 2018 that the average value of the national comprehensive financial development index dropped from 0.1539 to 0.1210, while the average value of the national green building comprehensive development index increased from 0.1244 to 0.1389. This shows that the development of green buildings in China’s provinces (autonomous regions and municipalities) lagging behind the development of the financial industry has undergone a significant change. The green transformation of China’s green buildings is increasingly supported by the financial industry. However, the development level of China’s financial industry has been suppressed in recent years, which may be caused by downward pressure on the economy.

#### 5.2.2. Empirical Analysis of the Degree of Development and Degree of Coupling of Green Buildings and Financial Industry in China

This paper divides the degree of development of green buildings and financial industry in China in 2014 and 2018 into five grades according to the natural discontinuous point classification method—namely, low degree of development (0.00000–0.10000), relatively low degree of development (0.10001–0.20000), medium degree of development (0.20001–0.30000), relatively high degree of development (0.30001–0.40000), and high degree of development (0.40001–0.50000), using ArcGIS 10.8 software to create the spatial distribution of the degree of development of the national financial industry and green buildings. As shown in Figure 2, in 2014 only Jiangsu was at a high level of development with a degree of 0.4311. Among them, the five economically developed provinces of Guangdong, Jiangsu, Zhejiang, Shandong, and Beijing are also significantly more developed than other regions and can be listed as the first cohort. Sichuan, Hubei, Henan, Hebei, and Liaoning have similar economic scales, and so there is no obvious gap between the development levels of these provinces (autonomous regions and municipalities), and they belong to the second cohort. The development statuses of green buildings and financial industries in Fujian, Hunan, Jiangxi, Anhui, Chongqing, and Shaanxi provinces and cities are all weaker than those of the second cohort, and so the degree of development is classed as the third cohort. Due to the influence of resource endowment and population size, Yunnan, Guizhou, Heilongjiang, Jilin, and other provinces have a low level of development and are in the fourth cohort. Hainan, Gansu, Tibet, and Qinghai are affected by geographical location, environmental climate, and other factors, resulting in them being in the fifth cohort of development. 

In 2018 Henan and Sichuan entered the first cohort, Fujian and Anhui entered the second cohort from the third cohort, the development levels of Beijing, Liaoning, and Tianjin declined significantly, and the development levels of other provinces did not change significantly. Overall, the degree of development of the central and eastern parts of China is relatively high, while that, overall, of green buildings and financial industry has shown a downward trend in recent years. The average degree of development has dropped from 0.1375 in 2014 to 0.1317 in 2018. The development gap between provinces (autonomous regions and municipalities) has not narrowed significantly.

The degree of coupling above between green buildings and the financial industry in China in 2014 and 2018 is divided into five levels by the natural discontinuity classification method, namely, low degree of coupling (0.000000–0.600000), relatively low degree of coupling (0.600001–0.700000), medium degree of coupling (0.600001–0.700000), relatively high degree of coupling (0.800001–0.980000), and high degree of coupling (0.980001–1.000000)—using ArcGIS10.8 software to create the financial industry and green spatial distribution map of buildings’ degree of coupling. As shown in Figure 3, in 2014, except for Ningxia, which has a high degree of coupling, other provinces in the northwest region are in the high degree of coupling stage. In North China, except for Heilongjiang, Beijing, and Hebei, which have a relatively high degree of coupling, all other provinces in North China have a high level of coupling. In South China, only Guangdong has a high degree of coupling, as other provinces are in the high degree of coupling stage. Sichuan and Tibet in the southwest are in the stage of high degree of coupling and low degree of coupling, respectively. In 2018 Sichuan, Beijing, and Hebei rose to a high level of coupling, and Tibet’s coupling rose from 0.1653 to 0.5668, or an increase of more than 3.4 times, while the coupling levels of Hubei, Chongqing, Jiangxi, and Fujian declined. Overall, China’s overall level of coupling between green buildings and the financial industry is relatively high. Except for Tibet, there is no obvious gap between other provinces (autonomous regions and municipalities directly under the central government). In addition, the degree of coupling increased slightly compared with 2014, with the average value rising from 0.9567 to 0.9648, while the degree of coupling difference between provinces (autonomous regions and municipalities) significantly fell.

#### 5.2.3. Empirical Analysis of the Degree of Coupling Coordination of Green Buildings and Financial Industry Development in China

Combining the degree of coupling coordination classification standard (Table 8) and the calculated green building-financial industry degree of coupling coordination of each province (autonomous region and municipality). As shown in Figure 4, only Jiangsu Province was situated in a good coupling and coordination range in 2014 and 2018, which is inseparable from the high level of financial development in Jiangsu Province and the active implementation of the green building “ecological cultural landscape” development concept. The provinces with intermediate coordination are mainly concentrated in the coastal areas, including Zhejiang, Guangdong, Beijing and Shandong. The financial systems of the economically developed provinces have strong risk resistance capabilities, and the volume of the real economy is sufficient to support the stable development of the financial industry. This provides a long-term guarantee for the development of green buildings. The provinces in primary coordination include Sichuan, Hebei, Liaoning, Henan, Hubei, etc., while the degree of coupling coordination in Ningxia, Gansu, and Inner Mongolia is at a serious level of imbalance. We see that the coordinated development of China’s financial industry and green buildings has a stepped effect in space. Compared with 2014, in 2018 only 10 provinces including Guizhou, Sichuan, Hubei, Hunan, and Fujian showed an upward trend in the degree of coupling and coordination, and the synergistic development effect was enhanced. The coupling coordination degree of the remaining 21 provinces and regions has declined. This shows that the financial policies of central China and some southwestern provinces are increasingly inclined toward investment in green buildings to avoid the impact of environmental risks on the financial industry, while the relevant financial support policies in other regions are still fragmented. During the research period, in terms of the spatial agglomeration trend of green buildings and regional economic development, the degree of coupling coordination of the two shows the spatial pattern characteristics of East China > Central China > South China > North China > Southwest China > Northwest China. The level of coupling coordination between green buildings and the financial industry is on the verge of coordination as a whole.

## 6. Discussion

Innovation solves the problem of development momentum, while green development solves the problem of harmonious coexistence between man and nature. This paper focuses on the problem of coupling and coordinated development between the financial industry and green buildings, makes an objective and reasonable evaluation of the level of coupling and coordinated development between the financial industry and green buildings in each province, and provides a reference for local government to formulate industrial policies. However, the coordinated development mechanism of the financial industry and green buildings is a multi-agent and multi-path interactive process, and its complexity cannot be explained effectively by traditional “linear thinking”. Therefore, on the basis of previous research, this paper determines the weights of 18 indicators through factor analysis, and effectively eliminates irrelevant factors in multi-dimensional correlated variables, reducing the complexity of the model. Relatively innovative development indices are constructed to evaluate the financial industry and green building of each province in China. Then, based on the financial industry–green building system coupling coordination degree model, the level of coordinated development of the two was successfully evaluated. The results show that the financial industry and green buildings in the eastern coastal area have high coupling, and the coupling and coordinated development has great potential.

This research has made a powerful supplementary exploration of the complex relationship between the financial industry and green buildings in developing countries from the perspective of coupled and coordinated development, and provided corresponding policy references and suggestions for the green transition of the traditional building industry. However, similar to other studies, this study also has some limitations. On the one hand, unfortunately limited by the length of the article, this paper only compares and analyzes the empirical results in 2014 and 2018 when constructing the relevant measurement data of the financial industry and green buildings in various provinces in China. However, from the results, in terms of the spatial agglomeration situation of China’s green buildings and regional economic development, the change trend of the coupling coordination degree presented by the financial industry and green buildings is not large, which does not affect the final conclusion. On the other hand, there may also be some limitations in the generalizability of the empirical results. When constructing the index to measure financial industry and green building development, the selected alternative indicators will inevitably have shortcomings, which may also have a certain impact on the final results. At the same time, the relevant conclusions and recommendations may not be applicable to developed countries and developing countries with low levels of green development. In the future, our research will include a larger sample size, cover developed countries, poor countries and more developing countries, and try to improve the precision of the measurement indicators, in order to better contribute to the research on the path and mechanism of the coordinated development of the financial industry and green buildings around the world.

## 7. Conclusions 

(1) Through time series analysis, it can be seen that the degree of coupling and coordination between green buildings and the financial industry in China is generally low, and the development of green buildings in most provinces lags behind their financial development. The situation is generally on the verge of unbalanced development and is in the transitional stage of primary coordinated development. Specifically, it is characterized by the spatial difference of “high in the east and low in the west”, but the development of green buildings in most provinces lags behind the development of finance. Moreover, the coupling and coordination state of central China are improving, while the fluctuations in the northeast and northwest regions are relatively large. This indicates a regional imbalance in the development of green buildings and the financial industry.

(2) Through spatial analysis, it can be seen that the space for the coordinated development of green buildings and the financial industry in China presents a step-like effect, and the degree of coupling and the coordination of the coupling systems of the provinces (autonomous regions and municipalities) vary greatly, decreasing from the eastern coastal areas to the central and western regions. In the eastern region, Jiangsu, Zhejiang, Guangdong, and Beijing are relatively high-coordinated areas spreading to the surrounding areas. The provinces (autonomous regions and municipalities) in the northeast and central regions are developing in a balanced manner. In the western region, Sichuan is the center of coordinated development, and its degree of coordination is relatively high. Based on the above conclusions, local government should pay attention to the coordination relationship between green buildings and the development of the financial industry, formulate a coordination mechanism between the two according to local conditions, and promote their interactive development. Specific recommendations are as follows.

(3) The eastern region should focus on improving its leading role in the surrounding areas and actively explore the construction mode of a green building–financial development system based on different backgrounds, which can be shared with the whole country through networks, conferences, and other modes, so as to provide national-level benefits. The region’s green building–financial development model has accumulated more practical samples. At the same time, relying on its coastal location advantages, the authorities can accelerate the application of foreign advanced scientific research achievements in the domestic market and give full play to their role in the construction–financial ecosystem, so as to promote the coordinated and healthy development of the financial industry and green buildings in the surrounding areas.

(4) The central region must further strengthen financial cooperation with the eastern region and continue to expand the advantages of its industrial scale relative to agglomeration. At the same time, local government should guide and support the advancement of the financial industry and green buildings through fiscal and taxation measures, land, and other policies, so as to steer them more towards the incentives of market-oriented green financial supply, and give full play to fiscal credit and taxation policies to leverage social capital. Realizing the complementarity between the development of the financial industry and green buildings is the region’s core direction.

(5) For the northeastern region and the western region, attention should be paid to the cultivation and introduction of industry talents, accelerating the formation of talent agglomeration and industrial agglomeration, speeding up technology circulation among construction enterprises, and forming a scale effect. Through the transmission of important national strategic documents, special policies, and implementation rules, the government can guide the actions of all stakeholders in the market to achieve the coordinated development of the financial and construction industries. At the same time, focus can be placed on cooperation and exchanges among construction enterprises, scientific research institutions, and financial institutions to promote the healthy and rapid development of green buildings and the financial industry.

## Figures and Tables

**Figure 1 ijerph-19-08685-f001:**
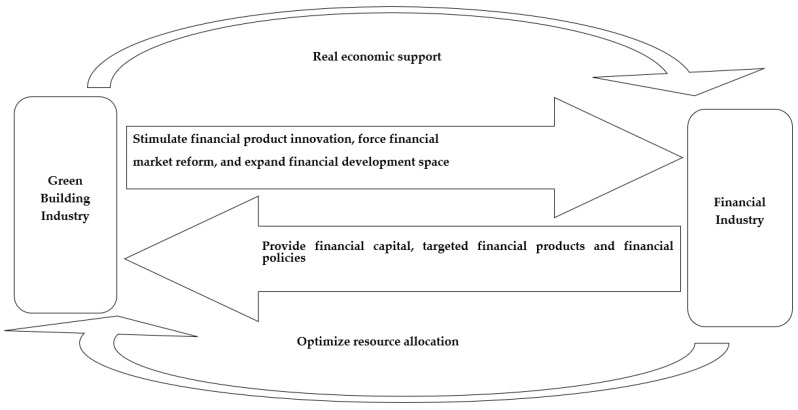
Road map for the coordinated development of green buildings and the financial industry.

**Figure 2 ijerph-19-08685-f002:**
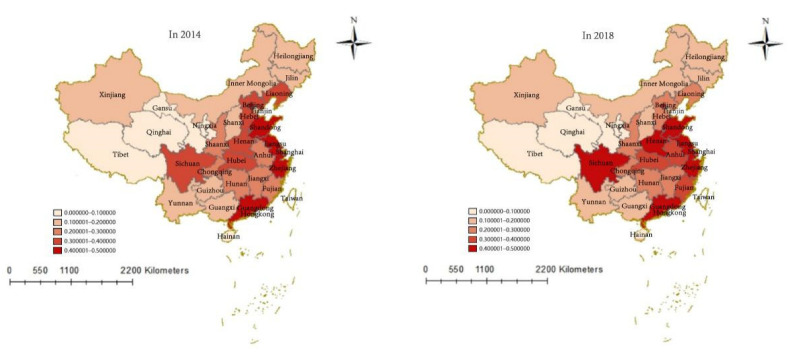
Spatial distribution in the degree of development of China’s financial industry and green buildings in 2014 and 2018.

**Figure 3 ijerph-19-08685-f003:**
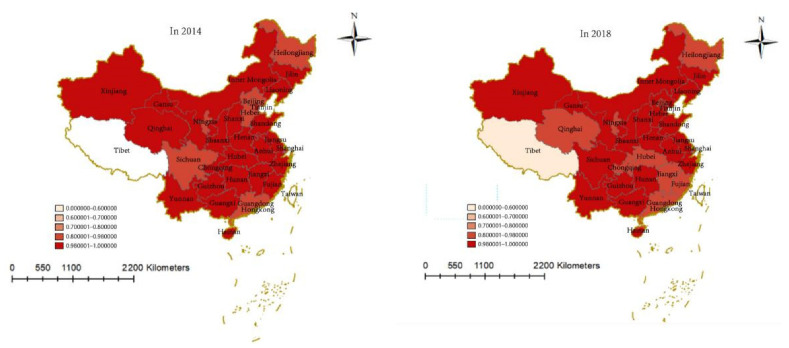
Spatial distribution of degree of coupling between China’s financial industry and green buildings in 2014 and 2018.

**Figure 4 ijerph-19-08685-f004:**
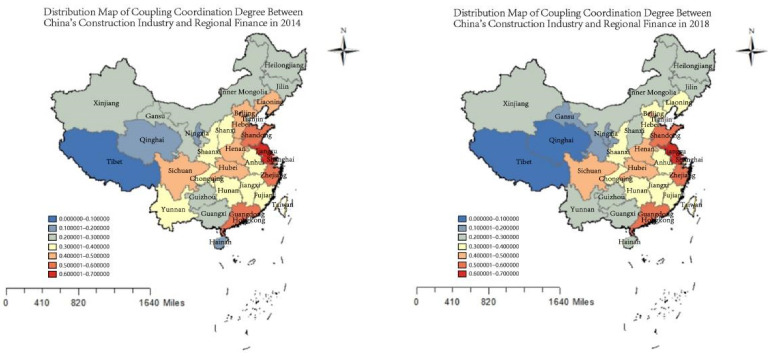
Spatial distribution of degree of coupling coordination between China’s financial industry and green buildings in 2014 and 2018.

**Table 1 ijerph-19-08685-t001:** Financial industry-regional green buildings’ green coupling coordinated development evaluation system.

Subsystem	Criterion Layer	Indicator Layer	Weight	
			2014	2018
Financial industry development	Financial scale	Total stock market value (100 million yuan)	0.04291	0.04319
		Scale of social financing (100 million yuan)	0.06268	0.06389
		Premium income (100 million yuan)	0.06268	0.06435
	Financial environment	Bank practitioners (units)	0.05564	0.04319
		Various deposits (100 million yuan)	0.06196	0.06389
		GDP (100 million yuan)	0.06092	0.06435
	Financial efficiency	Financial industry added value (100 million yuan)	0.06117	0.04319
		Total capital formation	0.05497	0.06389
		Total profit of ionstruction Industry (10,000 yuan)	0.06166	0.04319
	Industrial benefits	Gross output value of the construction industry (10,000 yuan)	0.05849	0.06389
		Contract amount of the construction industry (10,000 yuan)	0.06092	0.06435
		Industry scale, number of employed persons in the construction industry (persons)	0.05154	0.04319
Green building development level	Industrial scale	Completed area (10,000 square meters)	0.05247	0.06389
		Number of construction enterprise units (units)	0.05854	0.06435
	Ecological benefits	Ecological benefits, green space rate in built-up areas (%)	0.03818	0.04319
		Total environmental investment (100 million yuan)	0.04849	0.06389
		Number of construction environment documents (pieces)	0.04642	0.06435

**Table 2 ijerph-19-08685-t002:** Component matrix after rotation of observations in 2014.

Variable	Component 1	Component 2	Component 3
Stock market value	0.222	0.092	0.917
Social financing scale	0.696	0.380	0.567
Premium income	0.801	0.400	0.408
Banker	0.876	0.379	0.121
Various deposits of financial institutions	0.767	0.312	0.535
Gross domestic product	0.809	0.505	0.218
Value added in the financial industry	0.714	0.347	0.537
Gross capital formation	0.775	0.487	0.101
Urban and rural savings	0.872	0.351	0.306
Total green building profits	0.436	0.795	0.373
Gross output value of green buildings	0.403	0.872	0.225
Green building contract amount	0.465	0.760	0.355
Green building employment	0.329	0.921	0.050
Completed area	0.308	0.934	0.090
Number of construction companies	0.562	0.737	0.185
Green space rate in built-up area	0.070	0.258	0.769
Total environmental investment	0.203	0.517	0.603
Number of construction environment documents	0.885	0.229	0.007

Note: Our definition of the market value of stocks in each province is the market value of listed financial companies registered in the province (The following table defines the same).

**Table 3 ijerph-19-08685-t003:** Interpretation of the total variance of observations in 2014.

Element		Initial Eigenvalues			Rotational Load Sum of Squares	
	Total	Percent Variance	Grand Total %	Total	Percent Variance	Grand Total %
1	12.974	72.077	72.077	6.973	38.736	38.736
2	1.856	10.312	82.389	5.908	32.824	71.56
3	1.435	7.972	90.362	3.384	18.802	90.362
4	0.659	3.66	94.021			
5	0.405	2.249	96.27			
6	0.238	1.324	97.594			
7	0.136	0.754	98.348			
8	0.121	0.67	99.018			
9	0.047	0.263	99.281			
10	0.042	0.236	99.517			
11	0.033	0.186	99.702			
12	0.02	0.113	99.815			
13	0.012	0.067	99.882			
14	0.01	0.054	99.936			
15	0.007	0.039	99.976			
16	0.003	0.014	99.99			
17	0.001	0.005	99.995			
18	0.001	0.005	100			

**Table 4 ijerph-19-08685-t004:** Component matrix after rotation of observations in 2018.

Variable	Component 1	Component 2	Component 3
Stock market value	0.023	0.189	0.947
Social financing scale	0.530	0.461	0.650
Premium income	0.505	0.694	0.462
banker	0.452	0.792	0.291
Various deposits of financial institutions	0.335	0.573	0.723
Gross domestic product	0.583	0.677	0.409
Value added in the financial industry	0.397	0.529	0.693
Gross capital formation	0.043	0.343	0.029
Urban and rural savings	0.487	0.811	0.218
Total green building profits	0.801	0.311	0.375
Gross output value of green buildings	0.890	0.324	0.286
Green building contract amount	0.735	0.371	0.510
Green building employment	0.920	0.296	0.084
Completed area	0.937	0.248	0.138
Number of construction companies	0.742	0.560	0.199
Green space rate in built-up area	0.260	0.148	0.779
Total environmental investment	0.550	0.249	0.586
Number of construction environment documents	0.349	0.810	0.227

**Table 5 ijerph-19-08685-t005:** Interpretation of total variance of observations in 2018.

Element		Initial Eigenvalues			Rotational Load Sum of Squares	
	Total	Percent Variance	Grand Total %	Total	Percent Variance	Grand Total %
1	12.46	69.221	69.221	6.332	35.176	35.176
2	1.806	10.032	79.253	4.744	26.353	61.529
3	1.171	6.508	85.761	4.362	24.232	85.761
4	0.961	5.338	91.099			
5	0.658	3.655	94.755			
6	0.332	1.845	96.6			
7	0.17	0.946	97.545			
8	0.145	0.805	98.35			
9	0.092	0.514	98.864			
10	0.06	0.334	99.198			
11	0.054	0.298	99.496			
12	0.04	0.224	99.72			
13	0.019	0.104	99.824			
14	0.016	0.09	99.914			
15	0.007	0.041	99.955			
16	0.006	0.033	99.988			
17	0.002	0.009	99.997			
18	0.001	0.003	100			

**Table 6 ijerph-19-08685-t006:** Observation factor analysis KMO and the Bartlett test in 2014.

KMO Value		0.837
Bartlett’s test of sphericity	Approximate chi-square	1178.357
	Degrees of freedom	153
	*p* value	0.000

**Table 7 ijerph-19-08685-t007:** Observation factor analysis KMO and the Bartlett test in 2018.

KMO Value		0.807
Bartlett’s test of sphericity	Approximate chi-square	1049.937
	Degrees of freedom	153
	*p* value	0.000

**Table 8 ijerph-19-08685-t008:** Evaluation criteria for degree of coupling coordination.

Coupling Coordination	[0, 0.1]	(0.1, 0.2]	(0.2, 0.3]	(0.3, 0.4]	(0.4, 0.5]	(0.5, 0.6]	(0.6, 0.7]
Coordination level	Extremely out of balance	Severely disordered	Moderately disordered	On the verge of dysregulation	Primary coordination	Intermediate Coordinator	Well coordinated

**Table 9 ijerph-19-08685-t009:** Financial industry and green building sub-indices of provinces in China (Autonomous Regions and Municipalities) in 2014 and 2018.

Area	f_1_		f_2_		C		T		D	
	2014	2018	2014	2018	2014	2018	2014	2018	2014	2018
Beijing	0.3199	0.2475	0.1986	0.2228	0.9722	0.9986	0.2593	0.2351	0.5021	0.4846
Tianjin	0.1007	0.0616	0.0725	0.0610	0.9867	1.0000	0.0866	0.0613	0.2923	0.2476
Hebei	0.1984	0.1375	0.1313	0.1355	0.9791	1.0000	0.1648	0.1365	0.4017	0.3694
Shanxi	0.1058	0.0823	0.0810	0.0929	0.9911	0.9982	0.0934	0.0876	0.3042	0.2957
Inner Mongolia	0.0901	0.0586	0.0740	0.0540	0.9952	0.9992	0.0821	0.0563	0.2858	0.2372
Liaoning	0.1778	0.1192	0.1601	0.0978	0.9986	0.9951	0.1690	0.1085	0.4108	0.3286
Jilin	0.0856	0.0535	0.0639	0.0518	0.9894	0.9999	0.0747	0.0526	0.2719	0.2294
Heilongjiang	0.1005	0.0661	0.0571	0.0376	0.9612	0.9615	0.0788	0.0519	0.2752	0.2233
Shanghai	0.2408	0.1888	0.1134	0.1193	0.9331	0.9742	0.1771	0.1540	0.4065	0.3874
Jiangsu	0.4187	0.3432	0.4435	0.4589	0.9996	0.9895	0.4311	0.4011	0.6565	0.6300
Zhejiang	0.2903	0.2634	0.3479	0.3663	0.9959	0.9866	0.3191	0.3148	0.5637	0.5573
Anhui	0.1380	0.1222	0.1375	0.1734	1.0000	0.9849	0.1378	0.1478	0.3712	0.3815
Fujian	0.1464	0.1162	0.1372	0.2038	0.9995	0.9618	0.1418	0.1600	0.3765	0.3923
Jiangxi	0.1003	0.0888	0.1082	0.1575	0.9993	0.9604	0.1042	0.1232	0.3228	0.3439
Shandong	0.3291	0.2444	0.2329	0.3020	0.9853	0.9944	0.2810	0.2732	0.5262	0.5212
Henan	0.2258	0.1877	0.1591	0.2332	0.9849	0.9941	0.1924	0.2104	0.4353	0.4574
Hubei	0.1733	0.1437	0.1656	0.2191	0.9997	0.9781	0.1694	0.1814	0.4116	0.4212
Hunan	0.1436	0.1271	0.1226	0.1468	0.9969	0.9974	0.1331	0.1370	0.3642	0.3696
Guangdong	0.4899	0.4370	0.1987	0.2799	0.9062	0.9757	0.3443	0.3585	0.5586	0.5914
Guangxi	0.0948	0.0745	0.0676	0.0792	0.9859	0.9995	0.0812	0.0768	0.2829	0.2771
Hainan	0.0185	0.0782	0.0238	0.0222	0.9921	0.8301	0.0211	0.0502	0.1448	0.2041
Chongqing	0.1105	0.0856	0.1116	0.1307	1.0000	0.9781	0.1111	0.1082	0.3333	0.3253
Sichuan	0.2291	0.1954	0.1476	0.2264	0.9763	0.9973	0.1883	0.2109	0.4288	0.4586
Guizhou	0.0676	0.0604	0.0514	0.0715	0.9907	0.9964	0.0595	0.0660	0.2427	0.2564
Yunnan	0.0978	0.0713	0.0878	0.0988	0.9986	0.9869	0.0928	0.0850	0.3045	0.2897
Tibet	0.0001	0.0017	0.0136	0.0179	0.1653	0.5668	0.0068	0.0098	0.0336	0.0747
Shaanxi	0.1266	0.0938	0.0908	0.1106	0.9864	0.9966	0.1087	0.1022	0.3274	0.3191
Gansu	0.0562	0.0418	0.0384	0.0348	0.9822	0.9958	0.0473	0.0383	0.2156	0.1953
Qinghai	0.0134	0.0054	0.0144	0.0094	0.9993	0.9636	0.0139	0.0074	0.1178	0.0844
Ningxia	0.0121	0.0103	0.0295	0.0330	0.9081	0.8507	0.0208	0.0216	0.1375	0.1356
Xinjiang	0.0708	0.0507	0.0703	0.0584	1.0000	0.9975	0.0706	0.0546	0.2656	0.2333
minimum	0.0001	0.0017	0.0136	0.0094	0.1653	0.5668	0.0068	0.0074	0.0336	0.0747
maximum value	0.4899	0.4370	0.4435	0.4589	1.0000	1.0000	0.4311	0.4011	0.6565	0.6300
average value	0.1539	0.1244	0.1210	0.1389	0.9567	0.9648	0.1375	0.1317	0.3410	0.3330
standard deviation	0.1178	0.0994	0.0931	0.1089	0.1489	0.0835	0.1014	0.1009	0.1379	0.1389

## Data Availability

Not applicable.

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
