# Peer review of "The Coupling and Coordinated Development of Green Builds and Financial Development in China"

_ijerph, 2022, doi:10.3390/ijerph19148685_

Round 1
Reviewer 1 Report
One of the great weaknesses of the study is that it does not explain the reason for the selected years. Please indicate the reason why 2014 and 2018 are especially relevant years for the study. Why only 2 years and not choose 2015, 2016 and 2017?
On line 120, explain the consideration of nuclear energy as a green industry.
Delete lines 529 to 535 since they do not contribute anything in this section.
Author Response
Comments and Suggestions for Authors:
One of the great weaknesses of the study is that it does not explain the reason for the selected years. Please indicate the reason why 2014 and 2018 are especially relevant years for the study. Why only 2 years and not choose 2015, 2016 and 2017?
Reply:
We appreciate the reviewer’s insightful suggestion and the main reason why this article only compares the data in 2014 and 2018 is due to the length of the article. On the one hand, from the empirical results, the change trend of the degree of coupling and coordination between the financial industry and green buildings is not large, and it does not affect the final conclusion. By the way we have added this part of the explanation to the new Chapter 6 discussion.
On the other hand, in 2017, the Central Committee issued the "Opinions of the General Office of the State Council on Promoting the Sustainable and Healthy Development of the Construction Industry". Further deepen the reform of delegating power in my country’s construction industry, and speeding up industrial upgrading, so as to realize the efficient development of the construction industry. Driven by the concept of green development, green buildings will usher in a period of opportunity for multi-field integrated development. (Line 564-603, in red)
On line 120, explain the consideration of nuclear energy as a green industry.
Reply:
We feel sorry for the inconvenience brought to the reviewer. We think green energy in a narrow sense refers to renewable energy such as hydro, biomass, solar, wind, geothermal and ocean energy. These energy sources can be replenished after consumption with little pollution. Green energy in a broad sense includes the selection of energy sources with low or no pollution to the ecological environment, such as natural gas, clean coal and nuclear energy, in the process of energy production and consumption. In comparison, nuclear energy is much cleaner than other conventional energy sources. But it's true that, as you illuminatingly pointed out, this cleanliness is also relative, nuclear fuel is radioactive, and nuclear waste is difficult to handle, but we tend to choose a broad definition.
Delete lines 529 to 535 since they do not contribute anything in this section.
Reply: We thank the reviewer for pointing this out. We have removed these slightly redundant sentences
Reviewer 2 Report
Dear Authors,
I have carefully looked through your paper. Please, take my compliments on your work. I have some suggestions which may improve the quality of your article.
Abstract
There is no aim of the research.
Introduction
Some fundamentals points in that part are missing:
relation to the academic literature, research aim and research gap,
link to the theory,
research methodology,
expected results from the analysis,
structure of the paper
Literature review
there should be the hypothesis or research questions formulated at the end of that part
The impact of the financial industry on the development of green buildings
There are missing references to the literature
Selection of evaluation indicators and research methods
It is unclear what the expression means: stock market value of each province.
There are missing: sample selection and sample description, data collection and data description.
Conclusion and suggestions
No answer was found to the following questions:
What is the contribution to the research literature?
What is the limitation of the study?
What are the suggestions for future research?
Author Response
Comments and Suggestions for Authors:
Abstract:There is no aim of the research.
Reply: We greatly appreciate for your valuable comments as well as giving us the precious opportunity to revise the manuscript. We have revised the sentence as suggested to make it more reasonable. (Line 16-22, in red).
Introduction:
Some fundamentals points in that part are missing:
relation to the academic literature, research aim and research gap,
link to the theory,
research methodology,
expected results from the analysis,
structure of the paper
Reply: We are very grateful for your valuable comments and thank you so much for your careful check. We have carefully improved the content of the article to make it more complete and persuasive. (line110-134, in red)
Literature review
There should be the hypothesis or research questions formulated at the end of that part
Reply: Thanks to the reviewers for their criticism and corrections. The research of this paper is to construct the financial industry development index and green building development index of each province in China, and then use the coupling coordination model to empirically analyze the coupling and coordinated development level of the financial industry and green building in each province. But assumptions are not necessarily necessary in actual research. And we think that the research question has been explained in this paper (line 213-223, in red).
The impact of the financial industry on the development of green buildings. There are missing references to the literature
Reply: Thank you for your rigorous consideration. We have updated with new bibliographic references. (Line 259 and line 266 in red).
Selection of evaluation indicators and research methods
It is unclear what the expression means: stock market value of each province.
Reply: We thank the reviewer for pointing this out. We have added a new note below the table: The definition of stock market capitalization by province is the market capitalization of public financial companies registered in that province. (Line 370-371, in red).)
There are missing: sample selection and sample description, data collection and data description.
Reply: This article is limited by the length of the article, so there is no statistical representation of the sample, but we have explained the data source. (Line 337-340, in red).
Conclusion and suggestions no answer was found to the following questions:
What is the contribution to the research literature?
Reply: We apologize if our originals did not state what our research contribution was. We've added a new Chapter 6 discussion that addresses this issue. ( line 564-603,in red)
What is the limitation of the study?
Reply: We feel sorry if our originals did not state what our research contribution was. We've added a new Chapter 6 discussion that addresses this issue line 564-603, in red)
What are the suggestions for future research?
Reply: We feel embarrassed if our originals did not state what our research contribution was. We've added a new Chapter 6 discussion that addresses this issue line 564-603, in red)
Reviewer 3 Report
The paper has an interesting thesis to understand the "coupling" between the finance of green buildings and development.
However, the research needs to be more clearly articulated and explained for readers and researchers.
The calculations seem abstract are not related to the qualities of green buildings.
It is unclear how finance supports and receives value from green buildings compared to regular development.
Additional details and edits to provide more clarity of the research would help the paper.
Any refinement that moves the paper from abstract calculations to a rich and robust understanding of the actual coupling of finance and green buildings would be an improvement.
Do green buildings require a change in the value system of the financial industry? Are there key catalysts that have regions progress or move backward in the development of green buildings?
Author Response
Comments and Suggestions for Authors
The paper has an interesting thesis to understand the "coupling" between the finance of green buildings and development. However, the research needs to be more clearly articulated and explained for readers and researchers.
Reply: Thanks to the reviewers for their comments, this part is mentioned in the second chapter of the article. The mutual promotion between green building and finance is mainly reflected in the fact that capital investment promotes technological progress, and the scale of the industry can be further expanded. At the same time, compared with ordinary commodities, green building products have a long service life, a large amount of value storage, and stable value preservation, which will attract more capital injection, and the two will realize interconnection and interaction. Finance is the lubricant of the real economy, and the transformation of green buildings to green, as an emerging industry form, mainly practices the concept of sustainable development. On the one hand, green finance will guide funds into green environmental protection industries. At the same time, it has also continued to attract more participants to innovate and design green financial products, and actively guide the flow of capital into green industries such as wind power and nuclear energy. In addition, in this process, the main body is also encouraged to participate in the formation of the concept of green consumption and green production; on the other hand, the development of green industry pays attention to ecological environmental protection, and has obtained effective financing channels under the support of green finance. The sustainable development of green industry can be the development of green finance injects vitality. Specifically, green finance and green industry will interact in terms of factor supply, market orientation and related policies, thus forming a coupled development effect between the two.
The calculations seem abstract are not related to the qualities of green buildings.
Reply: Thanks to the reviewers for the confusion, we make the following answers. The domestic green building evaluation index has not yet formed a unified conclusion in the academic circle. Based on the existing literature research, this paper measures the green development level of the construction industry in each province (autonomous region and municipality) from three aspects: industrial scale, industrial benefit and ecological benefit. The total profit of the construction industry in the industrial benefit, the completed area and the number of construction enterprise units in the industrial scale, and the green space rate in the built-up area in the ecological benefit can all reflect the quality of green buildings.
It is unclear how finance supports and receives value from green buildings compared to regular development.
Reply: Thank you for your opinion. In the article, finance plays a boosting role in the process of economic and social development. Through reasonable financial policies, it can not only effectively reduce the cost of information transaction, but also improve the conversion rate of investment and savings, so as to promote the health of green buildings. steady development. First, the financial industry can promote the structural adjustment and optimization of the green building industry through the development of financial policies. As a real economy, green buildings need continuous support from financial policies in the process of development. In the process of financial policy implementation, the means of coordination and distribution can be used to continuously create rents. Therefore, the corresponding moral hazard and adverse selection can be avoided through incentives, and finally the purpose of promoting the upgrading of the green building industry structure can be achieved. Secondly, the financial market plays an important role in the supply and demand of green buildings, in order to provide high-quality financing services for green buildings. Because green buildings have their own characteristics in the development process, the capital required in the development of the industry is relatively large, and many projects need the support of financial policies to be carried out smoothly. Therefore, providing high-quality financial services and targeted financial products will have a profound impact on the development of green buildings. In the process of green building green transformation, financial services and financial policies need to be closely integrated, and ultimately form an interactive and mutually optimized relationship providing important financial policy support for the healthy and stable development of green buildings.
Additional details and edits to provide more clarity of the research would help the paper. Any refinement that moves the paper from abstract calculations to a rich and robust understanding of the actual coupling of finance and green buildings would be an improvement.
Do green buildings require a change in the value system of the financial industry?
Reply: We gratefully appreciate for your valuable comment. We will incorporate your inspiring considerations into future research plans.
Are there key catalysts that have regions progress or move backward in the development of green buildings?
Reply: Thanks to the reviewers for being kind enough to point out this error, and due to our omission, the key catalysts that have regions progress or move backward in the development of green buildings. Unfortunately, due to the limited length of the article, we will take your inspiring suggestions into consideration in future research plans